# RETHINKING LLM REASONING: FROM EXPLICIT TRAJECTORIES TO LATENT REPRESENTATIONS

**Cong Jiang**
Harbin Institute of Technology, Shenzhen
jerryjiang.jc@gmail.com

**Xiaofeng Zhang**
Harbin Institute of Technology, Shenzhen

**Fangzhi Zhu**
Independent Researcher

**Xiaowei Chen**
Independent Researcher

**Junxiong Zhu**
Independent Researcher

**Zheng Zhang**[*]
Harbin Institute of Technology, Shenzhen
Shenzhen Loop Area Institute, Shenzhen

## ABSTRACT

Large Language Models (LLMs) have achieved impressive performance on complex tasks by generating human-like, step-by-step rationales, referred to as *reasoning trajectory*, before arriving at final answers. However, the length of these reasoning trajectories often far exceeds that of the final answers, which incurs substantial inference costs even for relatively simple tasks. Advanced methods typically attempt to compress reasoning trajectory length through post-training, but they remain decoding-intensive and fail to inherently mitigate the efficiency challenge. In this work, we challenge the necessity of generating full reasoning trajectories and empirically demonstrate that LLMs can generate accurate answers using only fragmental reasoning paths, without relying on complete token-by-token sequences. To this end, we propose a novel **Latent Reasoning Tuning (LRT)** framework, which empowers LLMs to perform reasoning using implicit, compact, learnable representations instead of explicit textual trajectories. Technically, LRT replaces the costly autoregressive generation of reasoning steps with a single forward pass through a lightweight reasoning network, which generates latent vectors that encapsulate the necessary reasoning logic and condition the LLM to produce the final answer. Experiments on mathematical and out-of-domain benchmarks demonstrate that our LRT consistently outperforms relevant efficient reasoning methods. Moreover, by transforming explicit reasoning into latent reasoning, our approach surpasses the state-of-the-art Qwen3 hybrid reasoning framework. Code is available at https://github.com/MobiusDai/LRT.

## 1 INTRODUCTION

Recent advances in large language models (LLMs) have enabled slow-thinking reasoning models (Min et al., 2024), including OpenAI o1 (Jaech et al., 2024), DeepSeek-R1 (Guo et al., 2025), and Qwen QwQ (Team, 2025). The output of these models typically consists of a reasoning trajectory along with a summarized answer, the latter serving as a concise synthesis of the former. Through increased allocation of computational resources, these models have demonstrated significantly enhanced capabilities in solving complex tasks. Such reasoning capabilities are acquired through supervised fine-tuning (SFT) and reinforcement learning. For instance, DeepSeek-R1 employs Group Relative Policy Optimization (GRPO) with rule-based reward signals following the SFT phase, yielding models with superior reasoning performance.

Despite their impressive capabilities, reasoning LLMs often incur substantial computational overhead as they generate lengthy reasoning chains for backtracking and self-verification even for simple

---

[*]Correspondence to Zheng Zhang <darrenzz219@gmail.com>

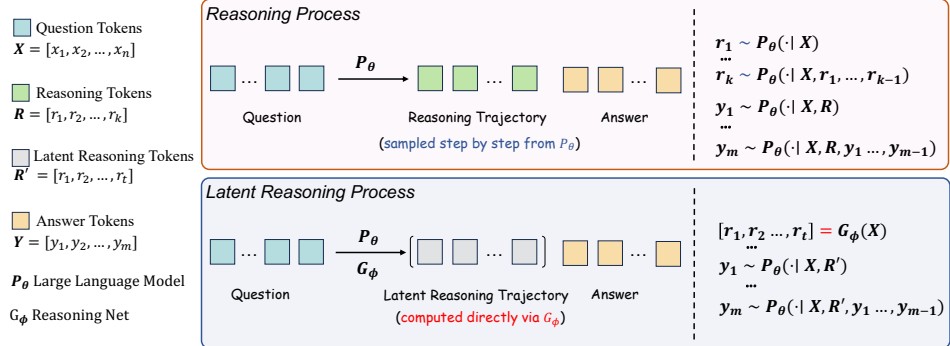

Figure 1: Comparison between the schematic diagrams of the reasoning LLM generation process and the Latent Reasoning method generation process.

tasks. This phenomenon, often referred to as *overthinking* (Sui et al., 2025), leads to lengthy reasoning chains that waste computation and increase inference latency, hindering real-time applications.

Mitigating the substantial inference costs of slow-thinking reasoning models, which are largely dominated by the auto-regressive generation of extended reasoning trajectories, has become a critical research imperative. A prominent research line explores post-training approaches that explicitly compress reasoning (Luo et al., 2025a; Hou et al., 2025). Methods such as ShorterBetter (Yi et al., 2025) construct dynamic rewards by selecting the shortest correct sample among multiple generations, while LC-R1 (Cheng et al., 2025) integrates collaboration-length and compression terms in addition to accuracy rewards. These reinforcement learning based approaches encourage shorter responses; however, they remain fundamentally "slow-thinking", the models still traverse extended reasoning trajectories, and the imposed length reward may even constrain problem-solving for real hard problems. A complementary line of work attempts to bypass reasoning entirely by substituting trajectories with fixed prompts. For instance, NoThinking (Ma et al., 2025) prefills a fabricated thinking block to skip chain-of-thought generation, and Qwen3 (Yang et al., 2025) enforces direct answer emission via a special control token. While such methods effectively eliminate reasoning tokens, their reliance on rigid prefilling introduces brittleness and can impair performance.

Unlike existing methods that pursue efficiency primarily via fine-tuning or prompt control, we propose latent reasoning tuning (LRT), a framework that fundamentally reimagines reasoning computation. As illustrated in Figure 1, we introduce an additional lightweight component, termed the *reasoning network*, to facilitate model reasoning. This component converts explicit reasoning trajectories into fixed-length, implicit latent representations, thereby obviating the need for autoregressive sampling of individual reasoning steps. The core of LRT is training reasoning network $G_\phi$ to generates latent reasoning chains which support the reasoning model to generate the final answers. Our approach mitigates the limitations of the two aforementioned methods: on the one hand, it replaces explicit reasoning trajectories with latent representations that can be directly computed; on the other hand, since our latent representations are derived from a reasoning network, they can be optimized to further enhance model performance, rather than relying on fixed representations for all inputs. Furthermore, its modular and non-intrusive design allows reasoning LLMs to be augmented without parameter modifications, thereby supporting seamless transitions between latent and explicit reasoning modes. Our primary contributions are as follows:

- We propose a novel Latent Reasoning Tuning framework which enhances reasoning efficiency by replacing the explicit, token-by-token generation of reasoning steps with a compact latent trajectory computed via an auxiliary network.

- The design of our framework is grounded in a key finding from our analysis of LLM reasoning. We demonstrate that models maintain high accuracy even when conditioned on fragmented reasoning trajectories, establishing that a fully explicit trajectory is not essential for correct inference. Building on this insight, our method further transforms the explicit trajectory into an latent representation.

- Experimental results demonstrate that LRT outperforms other efficient reasoning approaches when forcing the model to reason efficiently and surpasses the performance of Qwen3's non-thinking mode, thereby validating the effectiveness of our framework.

## 2 REASONING TRAJECTORY ANALYSIS

As analyzed in TokenSkip (Xia et al., 2025), tokens in the reasoning trajectory contribute unequally; many serve primarily as transitional elements that maintain coherence. These tokens can therefore be omitted to compress the trajectory, and token importance can be quantified using perplexity or a BERT-like language model. We further argue that, in slow-thinking models, the backtracking and self-verification behavior allows even certain important tokens or entire sub-steps to be partially compressed without loss of performance. To verify this hypothesis, we provide LLMs with reasoning trajectories of varying completeness and measure how these incomplete trajectories affected final answer accuracy.

**Settings.** Let $P_\theta$ be a reasoning LLM. In our implementation, we employ Deepseek-R1-Distill-Qwen-7B. Given a prompt $X$ and its corresponding reasoning trajectory $R$, the model defines a conditional distribution over answers $P_\theta(Y \mid [X, R])$. We can obtain the final answer by autoregressively sampling from this distribution $Y \sim P_\theta(\cdot \mid [X, R])$. To examine redundancy, we construct incomplete trajectories by randomly omitting certain tokens or steps and then compare the performance of models conditioned on complete reasoning trajectories with those conditioned on incomplete variants. The skipping scheme is designed at two levels of granularity:

- Token-level Skipping: For a skip rate $p \in [0, 1]$, construct $R_t(p)$ by independently deleting each token in $R$ with probability $p$ (preserving the order of the remaining tokens).

- Step-level Skipping: Segment $R$ into sentences/steps; for a skip rate $p \in [0, 1]$, construct $R_s(q)$ by randomly deleting each step with probability $p$.

We generate answers conditioned on these incomplete trajectories: $\hat{Y}_t(p) \sim P_\theta(\cdot \mid [X, R_t(p)])$ and $\hat{Y}_s(p) \sim P_\theta(\cdot \mid [X, R_s(p)])$.

**Observation.** Figure 2 presents a systematic comparison between models conditioned on complete reasoning trajectories and their incomplete variants across five skipping ratios. When provided with complete trajectories, the model consumes an average of 3529.3 tokens and achieves a pass rate of 92.8%, outperforming all incomplete counterparts. Notably, the performance degradation from trajectory ablation remains minimal. As the skip rate increases, model performance demonstrates remarkable robustness: when 30% of tokens are randomly omitted, the pass rate decreases by fewer than 2 percentage points; at a 50% token-level skip rate, the model maintains a 90.60% pass rate while utilizing approximately half the original trajectory length.

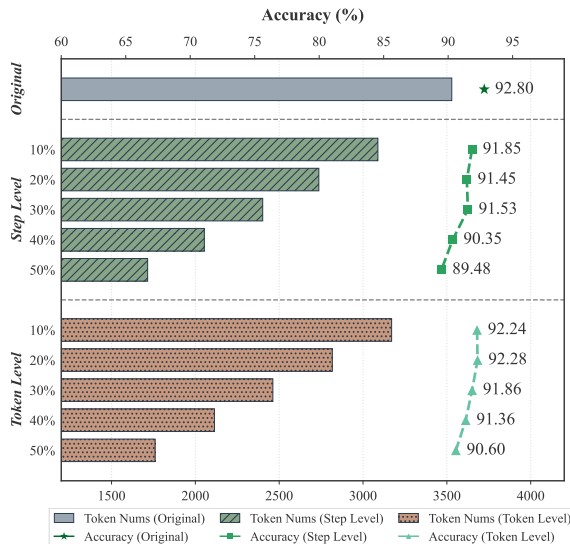

Figure 2: Experimental results of Deepseek-R1-Distill-Qwen-7B on Math-500 and corresponding token consumption in reasoning trajectories.

Based on the above observations, we can conclude that: **1. Reasoning trajectories exhibit substantial redundancy.** Consistent with our hypothesis, the model maintains robust performance despite skipping 50% of tokens or steps, demonstrating that reasoning trajectories contain significantly more information than required for correct answer inference. This suggests that current reasoning LLMs generate excessive intermediate representations. **2. Models demonstrate resilience to noisy or fragmental input.** The model is able to exploit salient information even from highly degraded trajectories, despite their higher perplexity. This robustness indicates that reasoning LLMs possess strong information-filtering capabilities and can identify critical reasoning components amid substantial noise or incompleteness.

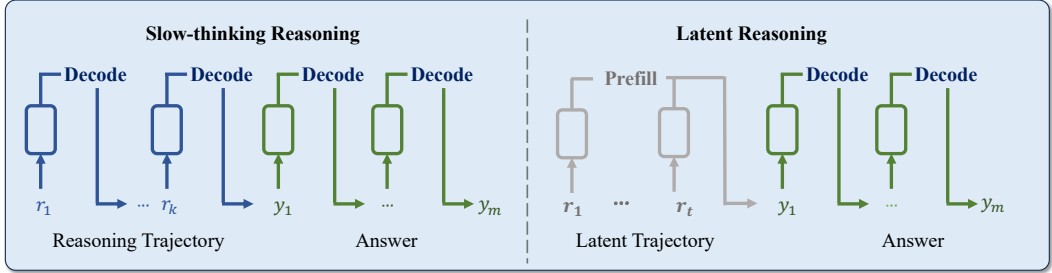

Figure 3: Explicit Slow-thinking Reasoning vs. Latent Reasoning. Comparison of the decoding process. Our latent reasoning performs the reasoning steps in compact latent representations, avoiding costly intermediate text generation.

## 3 METHOD

In this section, we begin by analyzing the reasoning model alongside existing efficient reasoning approaches. We then introduce our proposed latent reasoning tuning framework, elaborating on its architectural components, training methodology, and inference procedure.

### 3.1 PRELIMINARY

For a reasoning LLM $P_\theta$, the generation process typically involves producing intermediate reasoning content before arriving at the final answer to a given prompt. Formally, we denote the input prompt as $X = [x_0, \ldots, x_n]$, the reasoning trajectory as $R = [r_1, \ldots, r_k]$, and the final answer as $Y = [y_1, \ldots, y_m]$, where in general $k \gg m$. The reasoning trajectory is generated autoregressively according to the conditional distribution $P_\theta(\cdot \mid X)$. This process can be expressed as:

$$r_1 \sim P_\theta(\cdot \mid X), \quad r_2 \sim P_\theta(\cdot \mid [X, r_1]), \quad \ldots, \quad r_k \sim P_\theta(\cdot \mid [X, r_1, \ldots, r_{k-1}]), \qquad (1)$$

where $[\cdot, \cdot]$ denotes the concatenation operation. The final answer is also sampled in this way, can be expressed as: $Y \sim P_\theta(\cdot \mid [X, R])$.

Previous research has focused on internalizing the reasoning process by fine-tuning models to directly predict the final answer $Y$ without explicitly generating intermediate trajectories $R$. The optimization objective is to learn a model $P_{\hat\theta}$ such that $P_{\hat\theta}(\cdot \mid X)$ approximates the behavior of $P_\theta(\cdot \mid [X, R])$, effectively bypassing explicit reasoning. However, discarding intermediate reasoning steps often leads to suboptimal reasoning quality and reduced adaptability. Moreover, these methods typically output only the single answer, without any summary of the underlying rationale. Another line of work employs reinforcement learning to encourage models to generate more concise reasoning trajectories $\hat R$. These approaches optimize $P_{\hat\theta}$ such that $P_{\hat\theta}(\cdot \mid [X, \hat R])$ aligns with the original distribution while reducing redundancy in $R$. Nevertheless, even the shortened trajectories $\hat R$ remain considerably longer than the final answer $Y$, thereby limiting efficiency gains.

Moreover, both internalization-based and RL-based approaches require retraining the model, which substantially hinders the ability to leverage long-form reasoning for more challenging tasks.

### 3.2 LATENT REASONING TUNING

Our framework bypasses the computationally expensive process of generating explicit reasoning traces (illustrated in Figure 3). Instead, it utilizes compact latent representations, thereby eliminating the redundancy inherent in step-by-step reasoning.

Under greedy decoding, the generation of the reasoning trajectory becomes a deterministic process. We can therefore formalize this process (Equation 1) as a function $h : \mathcal{X} \times \Theta \to \mathcal{R}$, where $R = h(X, \theta)$. Consequently, the probability distribution for the final answer is given by:

$$P_\theta(Y \mid [X, R]) = P_\theta(Y \mid [X, h(X, \theta)]). \qquad (2)$$

---

**Algorithm 1** Latent Reasoning Tuning Framework

---

**Require:** Base model $P_\theta$, reference model $P_{\text{ref}}$, training dataset $\mathcal{D} = \{(X_i, Y_i)\}_{i=1}^N$
**Ensure:** Trained reasoning network $G_\phi$
 1: **Initialize:** Reasoning network $G_\phi$ with parameters $\phi$
 2: **Freeze:** Base model parameters $\theta$

 3: **Stage 1: Supervised Fine-tuning**
 4: **for** batch $(X, Y) \in \mathcal{D}$ **do**
 5:     $E_X \leftarrow \text{Embeddding}_\theta(X)$                    ▷ Get the embedding of input tokens
 6:     $H_X \leftarrow \text{HiddenStates}_\theta(E_X)$                    ▷ Extract final hidden states
 7:     $z \leftarrow G_\phi(H_X)$                    ▷ Generate latent reasoning
 8:     $\mathcal{L}_{SFT} \leftarrow -\log P_\theta(Y \mid [E_X, z])$                    ▷ Compute SFT loss
 9:     Update $\phi$ using $\nabla_\phi \mathcal{L}_{\text{SFT}}$                    ▷ Update reasoning network

10: **Stage 2: Reinforcement Learning**
11: **for** batch $X \in \mathcal{D}$ **do**
12:     $E_X \leftarrow \text{Embedding}_\theta(X)$                    ▷ Get the embedding of input tokens
13:     $H_X \leftarrow \text{HiddenStates}_\theta(E_X)$                    ▷ Extract final hidden states
14:     $z \leftarrow G_\phi(H_X)$                    ▷ Generte latent reasoning
15:     $\hat{Y}_{1:k} \leftarrow \text{Sample}(P_\theta(\cdot \mid [E_X, z]))$                    ▷ Generate $K$ candidate answers
16:     $r_{1:k} \leftarrow \{\text{ComputeReward}(\hat{Y}_k, Y)\}_{k=1}^K$                    ▷ Compute rewards
17:     $\bar{r} \leftarrow \text{mean}(r_{1:k}), \ \sigma_r \leftarrow \text{std}(r_{1:k})$
18:     $A_k \leftarrow (r_k - \bar{r})/\sigma_r$                    ▷ Normalized advantages
19:     $\rho_k \leftarrow \text{ComputeRatio}(P_\theta, P_{\text{ref}}, \hat{Y}_k)$
20:     $L_{GRPO} \leftarrow -\frac{1}{K}\sum_{k=1}^K \min\left(\rho_k A_k, \ \text{clip}(\rho_k, 1-\epsilon, 1+\epsilon)A_k\right)$                    ▷ Clipped policy loss
21:     Update $\phi$ using $\nabla_\phi \mathcal{L}_{\text{GRPO}}$                    ▷ Policy update

22: **Inference:**
23: **function** GENERATEANSWER($X_{test}$)
24:     $E_X \leftarrow \text{Embeddding}_\theta(X_{test})$
25:     $H_X \leftarrow \text{HiddenStates}_\theta(E_X)$
26:     $z_{test} \leftarrow G_\phi(H_X)$
27:     $Y_{pred} \leftarrow \text{Decode}(P_\theta(\cdot \mid [E_X, z_{test}]))$
28:     **return** $Y_{pred}$

---

The function $h$ preserves the autoregressive structure of reasoning, ensuring that each token $r_t \in R$ depends causally on its predecessors $r_{<t}$. However, our analysis in Section 2 demonstrates that complete step-by-step trajectories are not essential for achieving high performance. This finding indicates that LLMs primarily leverage salient trajectory components to derive final answers, suggesting that the strict autoregressive constraint is not a prerequisite. Motivated by this insight, we propose circumventing the explicit generation process by introducing a dedicated *reasoning network*, $G_\phi : \mathcal{X} \to \mathcal{Z}$, which directly maps inputs to compact latent representations of reasoning trajectory:

$$z = G_\phi(X). \tag{3}$$

The latent representation $z$ is optimized to support the downstream prediction of the correct answer, thereby serving as a compact surrogate for the explicit trajectory $R$.

To train the reasoning network $G_\phi$ to generate effective latent representations, we employ a two-stage training paradigm. The first stage uses Supervised Fine-Tuning (SFT) to align the behavior of the reasoning network with the reasoning LLM. The second stage leverages reinforcement learning to further enhance its problem-solving capabilities. The two-stage training and inference process is presented in Algorithm 1.

The primary objective of the SFT stage is to ensure that the latent trajectories produced by $G_\phi$ enable the reasoning model $P_\theta$ to replicate the answer of its original, explicit reasoning process. Formally,

we aim to make the conditional probability distribution $P_\theta(\cdot \mid [X, G_\phi(X)])$ closely approximate the target distribution $P_\theta(\cdot \mid [X, h(X, \theta)])$. A common approach for aligning distributions is knowledge distillation, which would involve minimizing the KL-divergence between them. But this method requires generating logits for the target distribution, a computationally prohibitive step. We therefore adopt a more direct and efficient SFT approach that circumvents this requirement. Our SFT dataset $\mathcal{D}$ consists of triplets $(X_i, R_i, Y_i)$ extracted from the outputs of a reasoning LLM, where $R_i$ denotes the reasoning trajectory and $Y_i$ the final answer. While the dataset contains both trajectories and answers, our training objective only leverages $(X_i, Y_i)$. For each input $X_i$, we first extract its final hidden state representation, $H_{X_i}$, from the reasoning model $P_\theta$. This state serves as a contextual embedding of the input $X_i$. The reasoning network $G_\phi$ then maps this representation to a latent trajectory. We optimize the parameters $\phi$ of the reasoning network by minimizing the negative log-likelihood, formally expressed as:

$$L(\phi) = -\log f_\theta(Y \mid [X, G_\phi(H_X)]). \tag{4}$$

While the first stage aligns the reasoning network with the reasoning model's behavior, it is inherently limited by the quality of the in the training data. To transcend this limitation and enhance the model's intrinsic problem-solving capabilities, we employ reinforcement learning in the second stage of training. In this stage, we refine $G_\phi$ by providing a direct reward signal based on the correctness of the final answer. This signal offers verifiable feedback for optimizing the latent reasoning process. Unlike SFT, which promotes imitation, the RL objective incentivizes the reasoning network to explore the latent space for more effective reasoning trajectories that consistently yield correct outcomes.

## 4 EXPERIMENTS

### 4.1 SETTINGS

**Models.** We evaluate our method alongside baseline approaches on DeepSeek-R1-Distill-Qwen-1.5B (Guo et al., 2025) and the Qwen3 series (Yang et al., 2025). DeepSeek-R1-Distill-Qwen-1.5B is a suitable model for evaluation as it is specifically optimized for reasoning tasks, making it a common target for efficiency improvements. The Qwen3 series features a native hybrid reasoning mode controlled via chat templates, where special tokens toggle between thinking and non-thinking modes. Our method offers an alternative approach to hybrid reasoning by transforming explicit reasoning models into latent reasoning models, achieving similar flexibility with improved efficiency. The reasoning network employs Qwen3-Embedding-0.6B (Zhang et al., 2025) to operate over a vocabulary of 256 learnable embeddings.

**Datasets.** For model training, we utilize the OpenR1-Math-220k dataset (Hugging Face, 2025) for supervised fine-tuning and the DeepScaleR-Preview-Dataset (Luo et al., 2025b) for reinforcement learning, respectively. To provide a comprehensive evaluation, we select five diverse reasoning benchmarks that encompass mathematical, logical, and scientific domains. Our assessment begins with GSM8K (Cobbe et al., 2021), which tests multi-step arithmetic reasoning through linguistically diverse grade-school word problems. For more advanced mathematical challenges, we utilize MATH-500 (Hendrycks et al., 2021; Lightman et al., 2023), a competition-level subset of the MATH dataset, alongside the American Mathematics Competitions (AMC) dataset (MAA, 2023), which assesses creative problem-solving and insight beyond routine calculations by demanding the synthesis of non-obvious solution strategies. To further probe the generalization capabilities of our method on out-of-domain benchmarks, we incorporate the LSAT (Zhong et al., 2023) and GPQA (Rein et al., 2024). The LSAT evaluates analytical deduction and reading comprehension through complex argumentative structures, targeting crucial abstract reasoning capabilities. Concurrently, GPQA gauges performance on expert-level scientific problems, presenting a collection of graduate-level questions in physics, chemistry, and biology specifically crafted to resist straightforward information retrieval and instead demand profound domain knowledge and intricate multi-step reasoning.

**Baselines.** In addition to the base model DeepSeek-R1-Distill-Qwen-1.5B, we compare our approach with several efficient reasoning methods: NoThinking (Ma et al., 2025), which bypasses the reasoning process through a simple prompt; ShorterBetter (Yi et al., 2025), which employs reinforcement learning with a dynamic reward signal to guide the model toward more efficient reasoning;

and LC-R1 (Cheng et al., 2025), which combines length and compression rewards to encourage the model to retain only the most critical steps.

## 4.2 RESULTS AND DISCUSSIONS

Table 1: Accuracy (%) of different baselines and our method on in-domain and out-of-domain tasks.

| Method | Budget | In-Domain Tasks | | | Out-of-Domain Tasks | | Average |
| --- | --- | --- | --- | --- | --- | --- | --- |
| | | AMC | MATH-500 | GSM8K | LSAT | GPQA | |
| Baseline | | 33.25 | 43.15 | 70.00 | 19.02 | 24.24 | 37.93 |
| NoThinking | | 37.75 | 58.35 | 73.24 | 18.15 | 23.74 | 42.25 |
| ShorterBetter | 512 | 33.87 | 55.11 | 60.78 | 19.05 | 26.23 | 39.01 |
| LC-R1 | | 35.75 | 48.00 | 74.26 | 18.59 | 24.24 | 40.17 |
| Ours | | **38.00** | **60.65** | **77.16** | **19.57** | **29.17** | **44.91** |
| Baseline | | 42.88 | 67.50 | **79.10** | 20.98 | 28.16 | 47.72 |
| NoThinking | | 40.25 | 66.70 | 75.00 | **22.72** | 25.88 | 46.11 |
| ShorterBetter | 1024 | 36.31 | 55.76 | 60.78 | 18.32 | 28.38 | 39.91 |
| LC-R1 | | **44.87** | 68.00 | 78.98 | 20.22 | **30.56** | 48.53 |
| Ours | | 42.50 | **68.50** | 78.95 | 22.39 | 30.55 | **48.58** |

**Comparison with Other Efficient Reasoning Methods.** Table 1 presents a comprehensive comparison between our method and four baseline models. Under the 512-token budget,, for in-domain tasks, our method improves baseline accuracy across three benchmarks from 33.25%, 43.15%, and 70.00% to 38.00%, 60.65%, and 77.16%, respectively. Compared to the NoThinking's prompt strategy, our method achieves an average improvement of 2.16%. Furthermore, our approach outperforms the RL-based efficient reasoning methods ShorterBetter and LC-R1 by average margins of 8.68% and 5.93%, respectively. For out-of-domain tasks, our method also demonstrates superior performance. Compared to the baseline model, it improves accuracy from 19.02% and 24.24% to 19.57% and 29.17%, respectively. In addition, our method surpasses the NoThinking, ShorterBetter, and LC-R1 approaches by average margins of 3.43%, 1.73%, and 2.96%, respectively. These results highlight the effectiveness of our method under limited token budgets. When more tokens are available, our approach continues to outperform other baselines in terms of average accuracy.

Table 2: Performance comparison of Qwen3 non-thinking mode and our method on Qwen3-1.7B and Qwen3-4B.

| Model | | In-Domain Tasks | | | Out-of-Domain Tasks | | Average |
| --- | --- | --- | --- | --- | --- | --- | --- |
| | | AMC | MATH-500 | GSM8K | GPQA | LSAT | |
| Qwen3-1.7B | base@1 | 44.50 | 66.05 | 66.79 | 30.81 | 26.52 | 46.93 |
| | ours@1 | 44.50 | 60.90 | 77.01 | 32.07 | 27.61 | **48.42** |
| | base@4 | 50.00 | 77.80 | 83.85 | 56.57 | 44.78 | 62.60 |
| | ours@4 | 51.00 | 77.40 | 89.61 | 62.12 | 53.91 | **66.81** |
| Qwen3-4B | base@1 | 47.25 | 70.70 | 75.08 | 44.82 | 32.50 | 54.07 |
| | ours@1 | 46.25 | 72.60 | 88.51 | 39.27 | 28.59 | **55.04** |
| | base@4 | 54.00 | 80.00 | 88.10 | 64.65 | 42.17 | 65.78 |
| | ours@4 | 54.00 | 84.80 | 95.07 | 67.17 | 56.96 | **71.60** |

**Comparison with the Qwen3 Series Models.** The Qwen3 series integrates thinking and non-thinking modes within a single model, with the mode determined by the chat template. Since our method transforms reasoning models into latent reasoning models, it offers an alternative approach for achieving hybrid reasoning. To evaluate this capability, we converted the thinking mode to latent reasoning and compared its performance against the corresponding non-thinking mode. We report results for Qwen3-1.7B and Qwen3-4B models enhanced with our method, evaluated against their non-thinking counterparts across five benchmarks. Performance is measured using the *pass@k*

metric: for each problem, $k$ samples are generated, and the problem is considered solved if at least one sample produces the correct answer.

As shown in Table 2, our Latent Reasoning Tuning method improves average accuracy across five benchmarks for both Qwen3-1.7B and Qwen3-4B models, increasing *pass@1* from 46.93% and 54.07% to 48.42% and 55.04%, respectively. The improvements in *pass@4* are even more substantial, rising from 62.60% and 65.78% to 66.81% and 71.60%, respectively. Notably, while *pass@1* performance occasionally matches or slightly underperforms the non-thinking mode, *pass@4* consistently surpasses it, indicating that our method generates more diverse solution paths. These results demonstrate the effectiveness of our approach in enabling hybrid reasoning capabilities.

## 4.3 ABLATION ANALYSIS

This section examines the results of our latent reasoning tuning methods with respect to the number of latent reasoning tokens and the training strategies employed.

Table 3: Accuracy (%) of the latent reasoning method with varying numbers of latent tokens.

| Tokens | In-Domain Tasks | | | Out-of-Domain Tasks | | Average |
|---|---|---|---|---|---|---|
| | AMC | MATH-500 | GSM8K | GPQA | LSAT | |
| 64 | 36.25 | 55.15 | 69.43 | 26.26 | 25.54 | 42.53 |
| 128 | 41.50 | 58.90 | 73.67 | 29.04 | 22.07 | 45.04 |
| 256 | 44.50 | 60.90 | 77.01 | 32.07 | 27.61 | **48.42** |
| 512 | 41.50 | 61.45 | 76.88 | 28.91 | 25.87 | 46.92 |

**Analysis of the Number of Reasoning Tokens.** Initially, we evaluated the LRM's performance using a fixed number of latent reasoning tokens. To further investigate the impact of token quantity, we conducted experiments with the Qwen3-1.7B model, varying the number of reasoning tokens from 64 to 512. All models were trained under identical experimental settings, with performance evaluated across five benchmarks. As shown in Table 3, performance improves as reasoning tokens increase up to $n \leq 256$, with accuracy rising from 42.53% to 45.04% and then to 48.42%. This finding aligns with the test-time scaling law, as performance improves with an increasing number of reasoning tokens. However, further increasing the token count does not yield continued improvements. When using 512 reasoning tokens, average performance falls below that of the 256-token model, with superior results observed only on the MATH-500 benchmark. This suggests that larger training scales may be necessary to fully leverage additional latent reasoning tokens.

Table 4: Accuracy (%) of the latent reasoning method under different training methods.

| Training Method | In-Domain Tasks | | | Out-of-Domain Tasks | | Average |
|---|---|---|---|---|---|---|
| | AMC | MATH-500 | GSM8K | GPQA | LSAT | |
| SFT | 37.00 | 54.65 | 63.64 | 28.66 | 22.50 | 41.29 |
| SFT + RL | 44.50 | 60.90 | 77.01 | 32.07 | 27.61 | **48.42** |

**Analysis of the training methods.** The reasoning network employs a two-stage training process. The first stage uses the reasoning dataset for supervised fine-tuning, followed by reinforcement learning to optimize the model and enhance its problem-solving capabilities. To evaluate the effectiveness of the two-stage approach, we compare the model trained solely with supervised fine-tuning to the model trained with both stages. As shown in Table 4, the two-stage training improves accuracy by 6.45%, 7.5%, and 13.37% on MATH-500, AMC, and GSM8K benchmarks, respectively. For out-of-domain tasks, we observe considerable improvements as well, with an average gain of 4.26%. These results demonstrate that the two-stage training strategy plays a vital role in enhancing problem-solving capabilities.

## 5 RELATED WORK

### 5.1 CHAIN-OF-THOUGHT REASONING

The reasoning capabilities of Large Language Models (LLMs) were significantly advanced by Chain-of-Thought (CoT) (Wei et al., 2022; Chu et al., 2023) prompting, a technique that elicits a sequence of intermediate steps to deconstruct complex problems before deriving a final answer. Building on this, a new paradigm of slow-thinking systems like OpenAI's o1 (Jaech et al., 2024), DeepSeek-R1 (Guo et al., 2025), and Qwen-QwQ (Team, 2025) has emerged to further elevate performance on challenging tasks (Li et al., 2025). These models employ extensive test-time computation, generating vast reasoning trajectories that are subsequently summarized into a concise answer. This process is typically optimized via Reinforcement Learning with Verifiable Rewards (RLVR) (Wen et al., 2025), where the model is rewarded based on the correctness of the final outcome. While highly effective, this slow-thinking paradigm is hampered by a critical inefficiency: *overthinking* (Sui et al., 2025). Models often generate disproportionately verbose reasoning chains, replete with redundant steps and superfluous calculations, particularly for simpler problems.

### 5.2 EFFICIENT REASONING

To mitigate the overthinking issue, prior work has explored several strategies to enhance the efficiency of reasoning LLMs (Feng et al., 2025). Some approaches focus on inference-time prompting, which guides models to generate more concise reasoning steps (Xu et al., 2025; Aytes et al., 2025) or even bypass reasoning entirely by forcing a direct answer (Ma et al., 2025). Other methods seek to instill this efficiency more directly by fine-tuning models on compressed reasoning chains. For instance, TokenSkip (Xia et al., 2025) and C3oT (Kang et al., 2025) utilize trajectories containing only keywords, while PAUSE Token (Goyal et al., 2023) replaces entire reasoning chains with special "pause tokens." ICoT-SI (Deng et al., 2024) internalizes reasoning chains through staged training on step-skipped datasets, enabling models to perform reasoning steps internally rather than explicitly. A particularly prominent strategy employs reinforcement learning to explicitly penalize verbosity, typically by incorporating a length-based penalty into the reward function. O1-Pruner (Luo et al., 2025a), for example, introduces a length-harmonizing reward, while other works apply penalties for exceeding token budgets (Aggarwal & Welleck, 2025; Hou et al., 2025) or use dynamic penalties based on answer correctness (Yeo et al., 2025). While effective, these methods still operate on explicit, token-based reasoning steps. Distinct from methods that shorten explicit reasoning, another line of work explores latent reasoning (Chen et al., 2025). These approaches (Hao et al., 2024; Saunshi et al., 2025; Wu et al., 2025; Geiping et al., 2025; Ruan et al., 2025) circumvent the generation of textual CoT steps by performing reasoning in a latent, continuous space. This computation is executed by iteratively refining the model's hidden states without decoding them into text at each step. Our approach diverges from these work by not training a latent reasoner from scratch. Instead, we adapt a pre-trained, explicit reasoning LLM, empowering it to leverage the latent representations for computation without generating intermediate text.

## 6 CONCLUSION

In this work, we investigate the overthinking problem by conditioning the answers of reasoning LLMs on fragmental reasoning trajectories. Our analysis shows that these models can exploit salient information even when trajectories are highly degraded and exhibit high perplexity. We then model the reasoning trajectory as a function of the input. Building on these insights, we introduce the Latent Reasoning Tuning (LRT) framework, which uses an auxiliary reasoning network to model the trajectory and encode it as a compact latent representation. With two-stage training, LRT effectively guides models toward correct reasoning. Comprehensive experiments across multiple benchmarks validate its effectiveness. Notably, since our method does not modify LLM parameters, it enables flexible switching between latent and explicit reasoning modes, offering a practical alternative to hybrid reasoning systems.

## ACKNOWLEDGMENTS

This research is partially supported by National Natural Science Foundation of China (Grant No. 62372132) and Shenzhen Science and Technology Program (Grant No. RCYX20221008092852077).

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

APPENDIX

## A  LLM USAGE STATEMENT

In the preparation of this manuscript, LLMs (Large Language Models) were employed exclusively for text polishing and grammar checking, with the goal of improving readability. All technical ideas, methodological designs, mathematical formulations, and experimental results were conceived, implemented, and verified solely by the authors.

## B  EXPERIMENT DETAILS

**Training Setup**. We train our latent reasoning network on eight NVIDIA A100 GPUs. The framework is implemented using the open-source TRL (von Werra et al., 2020) library. In the SFT stage, we train the reasoning network for 3 epochs with a batch size of 64 and a learning rate of $1 \times 10^{-3}$. In the subsequent RL stage, we use a batch size of 1024, generate 8 rollouts per question, and cap the maximum rollout length at 2048 tokens. RL training is performed for 100 steps with a learning rate of $1 \times 10^{-5}$ and a KL penalty coefficient of $2 \times 10^{-3}$.

**Inference Setup.** All inference experiments are conducted on a single NVIDIA A100 GPU to ensure a fair comparison of efficiency. We set the generation temperature to 0.6 and top-$p$ to 0.95. To evaluate performance specifically under efficient reasoning constraints, we adopt the budget-forcing implementation from S1 (Muennighoff et al., 2025) and enforce the same token budget across all models. Specifically, for the DeepSeek-R1-Distill-Qwen-1.5B model, we use 512- and 1024-token budgets, and for the Qwen3 series models, we use a 1024-token budget unless otherwise indicated in the tables. To ensure a strictly fair comparison of inference cost, all baselines and our method are evaluated using the same HuggingFace Transformers (Wolf et al., 2020) stack on identical hardware. Consequently, the reported speedups reflect algorithmic improvements rather than differences in system-level optimizations or implementation details.

## C  ARCHITECTURE OF THE REASONING NETWORK

As described in Section 4, the reasoning network $G_\phi$ is initialized from Qwen3-Embedding-0.6B (Zhang et al., 2025). Here, we provide additional architectural details. In our framework, the input to $G_\phi$ consists of the hidden states of the base model rather than token embeddings; accordingly, we remove the original embedding layer of $G_\phi$. We introduce a linear projection layer $f_{\text{in}}$ to map the base model's hidden states into the input space of $G_\phi$, and a second projection layer $f_{\text{out}}$ to project the output of $G_\phi$ back to the hidden dimension of the base model. The sequence length of the latent reasoning is determined by a set of learnable vectors $[\hat{r}_1, \hat{r}_2, \ldots, \hat{r}_t]$, which are optimized jointly during both the SFT and RL stages. Formally, the latent representations are produced as

$$z = f_{\text{out}} \left( G_\phi \left( f_{\text{in}}(H_X) \odot [\hat{r}_1, \hat{r}_2, \ldots, \hat{r}_t] \right) \right), \tag{5}$$

where $H_X$ denotes the hidden states of the base model for input $X$, and $\odot$ represents the Hadamard product with broadcasting. The reasoning network is not trained from scratch; instead, it is initialized from the pre-trained Qwen3-Embedding-0.6B model. We adopt this initialization because Qwen3-Embedding-0.6B has been trained on large-scale multilingual and long-text corpora, providing semantically rich representations and leading to substantially more stable training.

## D  COMPLEMENTARY EXPERIMENTS AND ANALYSIS

### D.1  COMPARISON WITH LARGER BASE MODEL

As shown in Table 5, on the Qwen3-8B backbone, our method achieves an average score of 59.63%, outperforming the non-thinking baseline by 3.19% on average. This demonstrates that our approach is not limited to 1.7B or 4B models and can effectively enhance the reasoning capabilities of larger 8B-scale base models.

To further illustrate the performance characteristics of our method, we also compare it with the thinking mode of the Qwen3-8B model. As shown in Table 5, the performance of Qwen3 (thinking

Table 5: Comparison of Qwen3 and our method on Qwen3-8B. The terms *non-thinking* and *thinking* denote the standard and reasoning modes, respectively.

| Method | Budget | In-Domain Tasks | | | Out-of-Domain Tasks | | Average |
|---|---|---|---|---|---|---|---|
| | | AMC | MATH-500 | GSM8K | GPQA | LSAT | |
| non-thinking | 1024 | 46.25 | 77.05 | 73.88 | 46.21 | 38.80 | 56.44 |
| ours | | 50.25 | 78.20 | 91.02 | 44.43 | 34.26 | **59.63** |
| thinking | 1024 | 48.50 | 71.30 | 90.52 | 40.15 | 29.57 | 56.01 |
| | 2048 | 51.25 | 83.50 | 90.54 | 48.23 | 39.67 | 62.64 |
| | 4096 | 55.75 | 90.10 | 91.04 | 56.82 | 54.35 | 69.61 |

mode) steadily improves as more token budget is allocated. When inference cost is not a constraint, the thinking mode can achieve even higher accuracy. However, in scenarios where low latency and computational efficiency are essential, our method provides a superior alternative.

## D.2 How Reasoning Tokens Interact with Base-Model Capacity

Table 6: Accuracy (%) of the latent reasoning method with varying numbers of latent tokens for larger base model (Qwen3-8B).

| Tokens | In-Domain Tasks | | | Out-of-Domain Tasks | | Average |
|---|---|---|---|---|---|---|
| | AMC | MATH-500 | GSM8K | GPQA | LSAT | |
| 256 | 50.25 | 78.20 | 91.02 | 44.43 | 34.26 | 59.63 |
| 512 | 50.75 | 78.50 | 92.49 | 44.95 | 33.15 | **59.97** |

As shown in Table 3, there exists a "sweet spot" in the number of latent tokens. For the Qwen3-1.7B base model, increasing the latent length from 256 to 512 results in a slight performance drop, indicating that 256 tokens already saturate the model's effective capacity. To further examine how model capacity interacts with latent-token length, we extend the analysis to a larger base model. As shown in Table 6, the Qwen3-8B model continues to benefit from longer latent trajectories within this range: increasing the length from 256 to 512 improves performance on four out of five benchmarks and yields a small gain in the overall average. These results suggest that larger base models possess sufficient capacity to exploit richer latent information, enabling the same 0.6B reasoner to leverage longer latent sequences (e.g., 512 tokens) and achieve higher performance. Consequently, the performance–length curve is expected to peak at a larger latent-token budget for larger models.

## D.3 Comparison of Inference Efficiency

Table 7: Performance comparison of inference cost on the Qwen3-1.7B model. The terms *non-thinking* and *thinking* refer to the standard and reasoning modes, respectively. The symbol † indicates that the computation accounts for the number of latent tokens.

| Method | Latency (sec/question) | Throughput (tokens/sec) | Peak Memory (MB) |
|---|---|---|---|
| thinking | 71.09 | 40.53 | 7538 |
| non-thinking | 14.62 | 48.93 | **3946** |
| ours | **11.79** | **51.31 (73.02$^{\dagger}$)** | 6528 |

To quantify the efficiency gains of the latent representations, we measured the average inference latency, throughput and peak memory on 64 random MATH-500 problems using the Qwen3-1.7B base model. We compare our method against the base model's thinking and non-thinking modes.

As shown in Table 7, our method achieves the lowest latency, even outperforming the non-thinking mode. This is because the reasoning network guides the base model to produce concise, direct

answers, thereby reducing the number of decoding steps. In terms of throughput, our method also delivers the highest effective throughput. Its standard token-level throughput is 51.31 tokens per second, and the effective throughput rises to 73.02 tokens per second when accounting for the 256 learned latent vectors processed in parallel. As for memory usage, the peak memory of our method falls between the non-thinking and thinking modes. This overhead is primarily attributable to the one-time generation of the latent representations, which temporarily increases activation and KV-cache usage and contributes to the higher throughput. After the latent representations are produced, decoding proceeds token-by-token as in the non-thinking mode, and memory consumption drops below this peak. These results confirm that our method delivers substantially higher information density per unit time while maintaining a memory footprint that remains well below that of slow-thinking reasoning.

### D.4 EMPIRICAL ANALYSIS OF LATENT REPRESENTATIONS

Table 8: Cosine similarity across latent representations of different benchmarks.

|  | AMC | MATH-500 | GSM8K | GPQA | LSAT |
|---|---|---|---|---|---|
| AMC | 0.438 | 0.565 | -0.173 | 0.104 | -0.276 |
| MATH-500 | 0.565 | 0.730 | -0.223 | 0.141 | -0.347 |
| GSM8K | -0.173 | -0.223 | 0.076 | -0.051 | 0.070 |
| GPQA | 0.104 | 0.141 | -0.051 | 0.149 | -0.032 |
| LSAT | -0.276 | -0.347 | 0.070 | -0.032 | 0.441 |

The latent representations are not linguistically interpretable in the way Chain-of-Thought traces are, since they are not composed of discrete tokens. While we have analyzed their functional role in Section 3 based on the training objective, here we further examine their empirical geometric structure across different benchmarks. For each dataset (AMC, MATH-500, GSM8K, GPQA, LSAT), we first average the latent representations over the sequence dimension to obtain a question-level latent vector. We then center these vectors by subtracting the global mean and compute the average pairwise cosine similarity between questions from any two benchmarks.

As shown in Table 8, three consistent patterns emerge: domain clustering, semantic separation, and complexity stratification. Competition-style math datasets (AMC and MATH-500) exhibit the highest cross-dataset similarity as well as strong within-domain similarity, suggesting that their latent representations are closely aligned despite covering different subdomains. LSAT exhibits positive within-domain similarity ($\approx 0.441$) but strongly negative similarity to AMC and MATH-500 ($-0.276$ and $-0.347$, respectively), indicating that logic-style reasoning occupies a distinctly different region of the latent space compared to olympiad-style mathematical reasoning. GSM8K and GPQA fall between these extremes: they show moderate self-similarity and relatively small cosine similarity with both competition math and LSAT, reflecting their hybrid reasoning characteristics.

Overall, these patterns indicate that the learned latent representations are organized by problem domain and difficulty, functioning as compressed, task-specific instructions and reasoning cues that guide the base reasoning LLM.

### D.5 STATISTICAL SIGNIFICANCE

Table 9: Mean $\pm$ standard deviation of accuracy across benchmarks for DeepSeek-R1-Distill-Qwen-1.5B under the 512-token budget.

| Benchmarks | AMC | MATH-500 | GSM8K | GPQA | LSAT |
|---|---|---|---|---|---|
| Mean Accuracy | 38.00 | 60.65 | 77.16 | 29.17 | 19.57 |
| Standard Deviation ($\sigma$) | $\pm 2.39$ | $\pm 1.39$ | $\pm 0.78$ | $\pm 1.40$ | $\pm 1.16$ |

As detailed in Section B, the answers are generated stochastically rather than deterministically. To mitigate the resulting variance, we already sample multiple reasoning paths for each query and report the average accuracy over these stochastic decodings. Thus, the improvements we report are not

artifacts of a single lucky sample, but reflect consistent performance gains. To further quantify the stability of our results and directly address the reviewer's concern regarding variability, we provide a detailed statistical breakdown (Mean $\pm$ Standard Deviation) in Table 9. As shown, most datasets exhibit very low variance ($\approx 0.8\%$–$1.4\%$), with AMC showing slightly higher variance due to its substantially smaller test set. In all cases, the performance improvements of our method exceed the corresponding standard deviations, indicating that the gains are robust rather than artifacts of stochastic sampling.

## E    DISCUSSION OF OTHER LATENT REASONING METHODS

Recent latent reasoning methods, including Coconut (Hao et al., 2024) and related work, share with our method the broad objective of reducing explicit chain-of-thought generation by operating in a latent space. The approaches, however, differ in mechanism, reasoning horizon, and architectural design. Existing methods often rely on iterative refinement of a recurrent hidden state, where the model repeatedly updates a continuous latent representation before producing the next token. This design encourages a local, stepwise form of latent computation (Zhu et al., 2025). Our method follows a different path. The reasoning network predicts an entire latent reasoning trajectory in a single forward computation, producing a sequence of latent vectors that represent the overall structure of the reasoning process rather than just the next refined state. This parallel formulation provides a larger expressive space and avoids entanglement with the base model's decoding loop.

The intended reasoning horizon also distinguishes the two approaches. Prior latent methods typically rely on only a small number of latent tokens, often fewer than ten, a scale suited to short-form inference or simple deductive steps. Our method is aimed at long-form reasoning tasks, where the explicit chain-of-thought becomes substantially longer. Our method allows the reasoning process to be expressed in parallel and reduces the tendency of the model to generate unnecessarily long or repetitive chains, thereby mitigating the overthinking problem.

A further distinction lies in how reasoning is integrated into the model architecture. Prior latent reasoning methods require retraining or substantial fine-tuning of the base LLM so that the model can internalize the latent computation. Our methods retains the base model entirely unchanged. The reasoning ability resides in a lightweight auxiliary module that can be enabled or disabled at inference time. This modular structure preserves the base model's original capabilities and allows seamless switching between latent reasoning and explicit chain-of-thought generation without modifying or reloading weights.

Overall, these points outline the key similarities and differences between our method and prior latent reasoning methods, offering a clearer view of its role in the broader development of latent reasoning techniques.

