# OpenReview forum: "Rethinking LLM Reasoning: From Explicit Trajectories to Latent Representations"
_ICLR.cc/2026/Conference — ICLR 2026 Poster_

### Official Review · Reviewer_gaFX · 2025-10-16

**Soundness:** 2
**Presentation:** 3
**Contribution:** 1
**Rating:** 4
**Confidence:** 4

**Summary:**

This paper introduces Latent Reasoning Tuning (LRT), a framework that replaces explicit token-by-token reasoning with compact latent representations generated by an auxiliary network. It aims to improve reasoning efficiency by performing implicit reasoning without generating lengthy step-by-step rationales.

**Strengths:**

+ The problem of overthinking and reasoning inefficiency is indeed practical and highly relevant for today’s LLMs, so it’s valuable to see it studied from this new latent-reasoning perspective.
+ The writing is clear and easy to follow—the method and its intuition are well-explained, making the technical parts digestible even on a first read.

**Weaknesses:**

+ Even though reasoning efficiency is an important problem, modern inference engines (like vLLM or SGLang) already accelerate long-token generation with KV-caching and routing optimizations. The proposed module doesn’t directly integrate with these frameworks, so it might actually reduce efficiency when generating the same number of tokens—this needs to be considered for a fair comparison with baselines.
+ Using only 512 tokens for distilled-R1 feels too artificial. For genuinely hard reasoning tasks (like AIME-style problems), original responses often exceed 10k tokens, so of course the performance drops drastically. It would make more sense to compare against short-response RL methods (like LC-RL, L1, or even RLVR with the same token cap) for a fairer setup.
+ The 512-token setup feels too synthetic and not representative of real-world reasoning scenarios. Trying longer contexts—even if not as long as full 32k setups—would strengthen the empirical validity.

**Questions:**

N/A

---

> ### Author Response · Authors · 2025-11-21
> **Response to Reviewer gaFX(1/3)**
>
> > 1 Even though reasoning efficiency is an important problem, modern inference engines (like vLLM or SGLang) already accelerate long-token generation with KV-caching and routing optimizations. The proposed module doesn’t directly integrate with these frameworks, so it might actually reduce efficiency when generating the same number of tokens—this needs to be considered for a fair comparison with baselines.
>
> **Response:**
> We thank the reviewer for this insightful comment regarding modern inference engines. We agree that frameworks like vLLM and SGLang significantly optimize throughput via PagedAttention and continuous batching. We address the comparison and efficiency concerns as follows:
>
> **For fair comparison:** To ensure a strictly fair comparison, all baselines and our LRT method in the paper were evaluated using the same standard HuggingFace Transformers[1] stack on identical hardware. Thus, the reported speedups are algorithmic and implementation-agnostic.
>
> **Importantly, LRT is compatible with vLLM/SGLang**: it is essentially a composition of a small LLM and a base LLM, both of which can benefit from the same engine-level optimizations (e.g., paged attention, continuous batching). Integrating LRT into vLLM/SGLang is therefore an orthogonal engineering improvement rather than a conceptual limitation of the method[2]. We are committed to releasing a vLLM-compatible implementation in our future open-source code to facilitate industrial adoption.
>
> [1] Transformers: State-of-the-Art Natural Language Processing. https://github.com/huggingface/transformers
>
> [2] https://docs.vllm.ai/en/v0.6.5/models/adding_model.html

---

> ### Author Response · Authors · 2025-11-21
> **Response to Reviewer gaFX(2/3)**
>
> > 2 Using only 512 tokens for distilled-R1 feels too artificial. For genuinely hard reasoning tasks (like AIME-style problems), original responses often exceed 10k tokens, so of course the performance drops drastically. It would make more sense to compare against short-response RL methods (like LC-RL, L1, or even RLVR with the same token cap) for a fairer setup.
>
> **Response:**
>
> Thanks for your insightful suggestion regarding the fairness of the token budget and the recommendation to further compare against short-response RL methods. We agree that restricting Distill-Qwen1.5B to 512 tokens might artificially cap its performance on harder tasks. To address this, we conducted a new set of experiments on DeepSeek-R1-Distill-Qwen1.5B with increased token budget. We relaxed the generation limit to 1024 tokens for all baselines.
>
> The results are presented in Table below:
>
> | Method         | AMC       | MATH-500  | GSM8K     | LSAT      | GPQA      | **Average** |
> | :------------- | :-------- | :-------- | :-------- | :-------- | :-------- | :---------- |
> | Baseline       | 42.88     | 67.50     | **79.10** | 20.98     | 28.16     | 47.72       |
> | NoThinking     | 40.25     | 66.70     | 75.00     | **22.72** | 25.88     | 46.11       |
> | ShorterBetter  | 36.31     | 55.76     | 60.78     | 18.32     | 28.38     | 39.91       |
> | LC-R1          | **44.87** | 68.00     | 78.98     | 20.22     | **30.56** | 48.53       |
> | **Ours (LRT)** | 42.50     | **68.50** | 78.95     | 22.39     | 30.55     | **48.58**   |
>
> **Analysis:**
>
> *   **Performance Gains:** As the reviewer hypothesized, increasing the budget to 1024 tokens did improve the Baseline and other methods’ performance compared to the 512-token setting.
> *   **LRT Robustness:** However, even against these enhanced baselines, LRT maintains the highest average performance (48.58).
>
> **Our original intent with the 512-token limit was to align the baselines with LRT's inference cost** (as LRT typically generates compact latent vectors followed by a short answer). The new results confirm that LRT maintains superior performance even when the token budget is extended to 1024 for all methods.

---

> ### Author Response · Authors · 2025-11-21
> **Response to Reviewer gaFX(3/3)**
>
> > 3 The 512-token setup feels too synthetic and not representative of real-world reasoning scenarios. Trying longer contexts—even if not as long as full 32k setups—would strengthen the empirical validity.
>
> **Response:**
>
> We appreciate the reviewer's suggestion to validate the method on longer contexts to better reflect real-world scenarios. To address this, we conducted a comprehensive evaluation using the **Qwen3-8B** backbone, testing token budgets from 1024 up to 4096 tokens.
>
> We present the detailed breakdown for both our  LRT method and the explicit thinking mode below:
>
> **Qwen3-8B(LRT)**
>
> | Token Budget | AMC   | MATH-500 | GSM8K | GPQA  | LSAT  | **Average** |
> | :----------- | :---- | :------- | :---- | :---- | :---- | :---------- |
> | 1024         | 50.25 | 78.20    | 91.02 | 44.43 | 34.26 | **59.63**   |
> | 2048         | 51.25 | 78.85    | 91.11 | 46.59 | 34.91 | **60.54**   |
> | 3072         | 51.50 | 80.25    | 91.17 | 46.48 | 34.04 | **60.69**   |
>
> **Qwen3-8B(thinking)**
>
> | Token Budget | AMC   | MATH-500 | GSM8K | GPQA  | LSAT  | **Average** |
> | :----------- | :---- | :------- | :---- | :---- | :---- | :---------- |
> | 1024         | 48.50 | 71.30    | 90.52 | 40.15 | 29.57 | **56.01**   |
> | 2048         | 51.25 | 83.50    | 90.54 | 48.23 | 39.67 | **62.64**   |
> | 3072         | 54.50 | 87.15    | 90.70 | 53.28 | 48.80 | **66.89**   |
> | 4096         | 55.75 | 90.10    | 91.04 | 56.82 | 54.35 | **69.61**   |
>
> **Analysis & Discussion:**
>
> **Efficiency at Practical Budgets:** As shown in the tables, **LRT significantly outperforms explicit Thinking at the 1024 token budget (59.63 vs. 56.01)**. This confirms that for latency-sensitive scenarios, LRT provides a much higher information density and reasoning capability per token.
>
> **Performance Saturation vs. Scaling:** We observe an interesting distinction in scaling behaviors:
>
> *   **LRT (Fast Reasoning):** Performance saturates quickly (improving marginally from 59.63 to 60.69 as budget triples). This is expected, as LRT is designed to compress reasoning into latent vectors and generate concise answers, typically completing its inference well within 1024 tokens.
> *   **Thinking Mode (Slow Reasoning):** Performance scales linearly with token budget, eventually surpassing LRT at higher budgets (e.g., 4096 tokens). This confirms that for extremely complex tasks requiring extensive search and self-correction, long-context generation is beneficial.
>
> **Modular Design:** These results highlight the strategic value of our framework's design choice to **keep the Base Model frozen**.
>
> *   LRT serves as a highly efficient **"Fast Thinking"** module for the majority of queries where speed and moderate cost are priorities.
> *   Since the Base Model remains intact, users retain the flexibility to switch to the **"Slow Thinking"** mode (explicit CoT) for complex problems that demand extensive computation (e.g., 4k+ tokens).
> *   Therefore, LRT is not a replacement for long-context reasoning but an efficient augmentation that optimizes the cost-accuracy trade-off for standard inference scenarios.

---

> ### Comment · Reviewer_gaFX · 2025-11-21
>
> OK, thanks for your experiments! I think it would be better to clarify the scenario distinction of Fast Thinking v.s. Slow Thinking. As you said, your method is mostly for the Fast Thinking scenario, and it would be better to just use the original model's thinking when the scenario allows for a higher token budget (Slow Thinking). It is important to make the appropriate scenario for your method clear to avoid overclaiming.
>
> I also increased my score.

---

> > ### Author Response · Authors · 2025-11-22
> > **Official Comment by Authors**
> >
> > We sincerely thank the reviewer for the positive feedback and for kindly raising the score.
> >
> > We fully agree that clearly distinguishing the “fast thinking” (latency constrained) regime from the “slow thinking” (high-budget long CoT) regime is important to avoid overclaiming. In the revised version, **we will explicitly position LRT as a fast-thinking framework that complements, rather than replaces, the original model’s long CoT capabilities,** and we will clarify this scenario distinction in the introduction, and discussion/limitations sections.

---

### Official Review · Reviewer_RA4D · 2025-10-30

**Soundness:** 3
**Presentation:** 2
**Contribution:** 2
**Rating:** 4
**Confidence:** 3

**Summary:**

This paper proposes Latent Reasoning Tuning (LRT), which replaces the explicit, token-by-token generation of reasoning trajectories in LLMs with compact latent representations. The authors show that reasoning LLMs can maintain high accuracy even when conditioned on fragmented reasoning paths, suggesting significant redundancy in explicit reasoning chains. LRT introduces a lightweight "reasoning network" G_phi that maps input questions to fixed-length latent trajectories, which then condition the base LLM to generate final answers.

**Strengths:**

The trajectory analysis provides compelling evidence that models are robust to token omission with minimal performance degradation.
It replaces O(k) autoregressive steps with a lightweight model with fixed reasoning size, which should provide efficiency gains.
The approach is modular, allowing switching between latent and explicit reasoning modes.
The results seem to show improvements over "baseline efficient reasoning" methods across several benchmarks.

**Weaknesses:**

Architecture of G is underspecified: I think there is no clear description of the reasoning network architecture. The paper mentions it uses "Qwen3-Embedding-0.6B" but it's unclear what the actual architecture is. The discussion says the latent reasoner is not trained from scratch, then it means it reuses some LLM parts? This was not clear to me when reading the paper.

Missing efficiency analysis: One of the main motivations for compressing reasoning chains is computational efficiency, but the computational overhead of the latent reasoning is not discussed. No metrics are provided for inference time, FLOPs, or memory usage. Since the approach uses fixed reasoning length and lightweight modules, there should be efficiency gains, but these are not reported or analyzed. It could be a useful addition to the paper.

No statistical significance testing: All tables lack measures of variability (confidence intervals or standard deviations). This makes it unclear whether and when the improvements are statistically significant. This is important for rigorous science.

Concurrent work: The paper mentions other latent reasoning work (Hao et al., 2024; Saunshi et al., 2025; Wu et al., 2025) in only one sentence of the related work section. It is important to better distinguish this work from other reasoning in latent space approaches. Even though most of them are fairly recent, the authors are aware of them and the discussion dismisses these related work too easily.

**Questions:**

What is the exact architecture of the reasoning network G?
What are the actual efficiency gains at inference time?
Are the performance improvements statistically significant? Can you provide confidence intervals or significance tests?

---

> ### Author Response · Authors · 2025-11-21
> **Response to Reviewer RA4D(1/4)**
>
> > 1 Architecture of G is underspecified: I think there is no clear description of the reasoning network architecture. The paper mentions it uses "Qwen3-Embedding-0.6B" but it's unclear what the actual architecture is. The discussion says the latent reasoner is not trained from scratch, then it means it reuses some LLM parts? This was not clear to me when reading the paper.
>
> **Response:**
> We agree that the current draft does not clearly describe the architecture of the reasoning network $G_\phi$; we clarify its architectural and initialization details below.
>
> 1. **Architecture.** **The reasoning network $G_\phi$ is initialized from Qwen3-Embedding-0.6B[1] which follows the standard Transformer architecture used in the Qwen3 series.** In our LRT framework, the input to $G_\phi$ corresponds to the hidden states of the base model rather than token embeddings; thus, we discard the original embedding layer of $G_\phi$. We introduce a linear projection layer $f_{\text{in}}$ to map the base model's hidden states to the dimensionality of $G_\phi$, and another projection layer $f_{\text{out}}$ to map the output of $G_\phi$ back to the base model’s hidden dimension. The sequence length of the latent reasoning is determined by a set of learnable vectors $[\hat r_1, \hat r_2, \ldots, \hat r_t]$, which are optimized jointly during the SFT and RL stages. Formally, the latent tokens are generated as:
>    $$
>    \text{latent tokens} = f_{\text{out}}(G_\phi(f_{\text{in}}(H_X) \odot [\hat r_1, \hat r_2, \ldots, \hat r_t])),
>    $$
>    where $H_X$ denotes the hidden states of the base model given input $X$, and $\odot$ denotes the Hadamard product (with broadcasting).
>
> 2. **Initialization.** **The reasoning network $G_\phi$ is not trained from scratch.** Instead, it is initialized using the pre-trained weights of Qwen3-Embedding-0.6B. We adopt Qwen3-Embedding-0.6B because it has been pre-trained on large-scale text data (covering multilingual and long-text understanding), which provides semantically rich representations and ensures more stable training compared to random initialization.
>
> We will incorporate these architectural and initialization details into the revised version of the paper.
>
> [1] Zhang et al. Qwen3 Embedding: Advancing Text Embedding and Reranking Through Foundation Models. arXiv preprint arXiv:2506.05176.

---

> ### Author Response · Authors · 2025-11-21
> **Response to Reviewer RA4D(2/4)**
>
> > 2 Missing efficiency analysis: One of the main motivations for compressing reasoning chains is computational efficiency, but the computational overhead of the latent reasoning is not discussed. No metrics are provided for inference time, FLOPs, or memory usage. Since the approach uses fixed reasoning length and lightweight modules, there should be efficiency gains, but these are not reported or analyzed. It could be a useful addition to the paper.
>
> **Response:**
> We appreciate the reviewer’s constructive suggestion regarding efficiency analysis. You are absolutely correct that replacing $O(k)$ autoregressive reasoning steps with a lightweight module and fixed-length latent representations leads to significant efficiency gains.
>
> To quantify this, we measured the average inference latency and throughput on 64 random MATH-500 problems using the Qwen3-1.7B base model. We compare LRT against the base model's explicit reasoning (thinking) and direct answering (non-thinking) modes.
>
> | Method                    | Latency (sec/question) |   Throughput (tokens/sec)   | Peak Memory (MB) |
> | :------------------------ | :--------------------: | :-------------------------: | :--------------: |
> | Qwen3-1.7B (thinking)     |         71.09          |            40.53            |       7538       |
> | Qwen3-1.7B (non-thinking) |         14.62          |            48.93            |     **3946**     |
> | **Qwen3-1.7B (LRT)**      |       **11.79**        | **51.31 (73.02$^\dagger$)** |       6528       |
>
> Key observations from the results:
>
> 1.  **Latency reduction:** LRT achieves the **lowest latency** (11.79s), representing a ~6x speedup compared to the Qwen3 thinking mode (71.09s). Notably, it even outperforms the non-thinking mode (14.62s). This is because the reasoning network \(G_\phi\) guides the base model to generate **more concise and direct answers**, reducing the total number of decoding steps compared to the non-thinking mode, and the reasoning network itself requires only a single forward pass, incurring minimal computational overhead.
> 2.  **Throughput gain:** LRT exhibits the highest effective throughput. The value **51.31** is the standard token-level throughput computed over the model’s output tokens. The value **73.02$^\dagger$** represents the effective throughput when additionally counting the 256 learned latent vectors as "reasoning steps" processed in parallel. This demonstrates that LRT achieves a much higher information density per unit of time compared to standard token-by-token generation.
> 3.  **Memory Usage:** LRT’s peak memory (6528 MB) lies between non-thinking and full CoT. LRT’s peak memory is higher than the non-thinking mode (3946 MB) because, during the latent reasoning stage, the 256 latent vectors are processed in parallel, which temporarily increases the activation/KV-cache footprint and boosts throughput. After the latent tokens are generated, decoding proceeds token-by-token as in the non-thinking mode, and the memory usage drops below this peak.
>
> We will include these metrics in the revised paper to provide a concrete analysis of the computational overhead and efficiency gains.

---

> ### Author Response · Authors · 2025-11-21
> **Response to Reviewer RA4D(3/4)**
>
> > 3 No statistical significance testing: All tables lack measures of variability (confidence intervals or standard deviations). This makes it unclear whether and when the improvements are statistically significant. This is important for rigorous science.
>
> **Response:**
> We fully agree with the reviewer on the importance of statistical rigor, particularly given the inherent stochasticity of reasoning models.
>
> We acknowledge that results based on single-run decoding can exhibit high variance. To mitigate this and ensure robust evaluation, in our experiments we already sample multiple reasoning paths (at temperature (T > 0)) for each query and report the **average accuracy** over these stochastic decodings. By averaging the performance over multiple stochastic samples across the entire dataset, this evaluation protocol substantially reduces the variance of the point estimates and provides a stable estimate of the model’s expected performance. Thus, the reported improvements are not due to a single lucky run, but reflect a consistent capability gain.
>
> To explicitly quantify this stability and address the concern regarding variability, we present the detailed statistical breakdown (Mean $\pm$ Standard Deviation) for our experimental results(Table 1) below:
>
> | Benchmarks                        | AMC        | MATH-500   | GSM8K      | GPQA       | LSAT       |
> | :-------------------------------- | :--------- | :--------- | :--------- | :--------- | :--------- |
> | **Mean Accuracy**                 | $38.00$    | $60.65$    | $77.16$    | $29.17$    | $19.57$    |
> | **Standard Deviation ($\sigma$)** | $\pm 2.39$ | $\pm 1.39$ | $\pm 0.78$ | $\pm 1.40$ | $\pm 1.16$ |
>
> **Analysis:**
>
> *   **High Stability:** For most datasets (MATH-500, GSM8K, LSAT), the standard deviations are very low, indicating that the performance of our method is highly consistent.
> *   **Sample Size Effect:** We observe slightly higher variance on AMC ($2.39\%$), which is statistically expected given its significantly smaller test set size ($N=100$) compared to others like GSM8K ($N=1319$).
> *   **Significance:** In all cases, the performance improvements of our method over the baselines (as reported in the main paper) exceed these standard deviations, confirming that the gains are genuine and statistically significant.

---

> ### Author Response · Authors · 2025-11-21
> **Response to Reviewer RA4D(4/4)**
>
> > 4 Concurrent work: The paper mentions other latent reasoning work (Hao et al., 2024; Saunshi et al., 2025; Wu et al., 2025) in only one sentence of the related work section. It is important to better distinguish this work from other reasoning in latent space approaches. Even though most of them are fairly recent, the authors are aware of them and the discussion dismisses these related work too easily.
>
> **Response:**
>
> We appreciate the reviewer pointing this out. We agree that the distinction between LRT and recent latent reasoning works (e.g., Hao et al., 2024) deserves a more in-depth discussion. While we share the goal of moving reasoning to a latent space, our approach differs fundamentally in **mechanism, scope, and modularity**.
>
> 1.  **Mechanism: Recurrent Refinement vs. Parallel Trajectory.**
>     **Methods like *Coconut*[1] typically rely on recurrent hidden states**, where the model loops over a continuous embedding to iteratively refine its "thought" before generating the next token. In contrast, LRT does not rely on iterative refinement of a single state. **Instead, our separate reasoning network predicts a comprehensive reasoning trajectory** (a sequence of e.g., 256 latent vectors) in parallel (or via a single forward pass of the reasoner). This trajectory encodes the entire reasoning path structure rather than just a refined next-step representation.
>
> 2.  **Scope: Short-Horizon vs. Long-Horizon.**
>     Existing latent approaches often utilize a very small number of latent tokens (typically fewer than 10) to replace Short-CoT or simple reasoning steps. In contrast, **LRT is designed to tackle complex, "slow-thinking" problems** (e.g., MATH benchmarks) where models typically suffer from "overthinking" and generate extremely long chains. We utilize a larger latent capacity (e.g., 256 latent tokens) to compress these long explicit chains, enabling the model to handle heavy computational logic that short latent bottlenecks cannot capture.
>
> 3.  **Modularity: Retraining vs. Plug-and-Play.**
>     Most prior works require **re-training or heavily fine-tuning of the base LLM** to adapt it to latent reasoning, which permanently alters the model's weights and behavior. Conversely, **LRT keeps the base LLM completely frozen**. The reasoning capability is encapsulated in the lightweight auxiliary module. This modular design not only preserves the base model's original capabilities but also allows for flexible switching between "Latent Reasoning Mode" (for efficiency) and "Explicit Thinking Mode" (for detailed reasoning trajectory) at runtime without reloading weights.
>
> We will incorporate this detailed comparison into the revised manuscript to clearly position LRT within the landscape of latent reasoning research.
>
> [1] Hao et al. Training Large Language Models to Reason in a Continuous Latent Space. https://arxiv.org/abs/2412.06769

---

> ### Author Response · Authors · 2025-11-24
> **Inquiry Regarding Rebuttal Feedback**
>
> Dear Reviewer RA4D,
>
> We sincerely appreciate the time and effort you have dedicated to reviewing our manuscript. We provided detailed responses to your comments a few days ago and wanted to check if you had any further questions. We remain fully available to address any additional concerns you may have.

---

> > ### Comment · Reviewer_RA4D · 2025-11-24
> >
> > Thank you for your detailed answer.
> >
> > I believe several points have been clarified and, if the authors do implement the modifications they discussed (especially regarding the latency, statistical testing, and discussion of related work), the paper could be in a publishable state. Therefore, I am raising my score.

---

> > > ### Author Response · Authors · 2025-11-24
> > >
> > > We sincerely appreciate your constructive feedback and are grateful for your decision to raise your score. We will incorporate the discussed revisions into the final manuscript. Thank you again for helping us improve our paper.

---

### Official Review · Reviewer_dCpV · 2025-10-31

**Soundness:** 4
**Presentation:** 4
**Contribution:** 4
**Rating:** 8
**Confidence:** 4

**Summary:**

This paper proposes a novel reasoning framework, latent reasoning tuning (LRT), which uses an auxiliary network to generate a sequence of latent representations in a single forward pass, and then concatenates the prompt and the latent representations to generate the final answer. Experimental results on various benchmarks show its consistent performance gain over baseline methods, showing its effectiveness.

**Strengths:**

1. The idea of using latent representations for efficient reasoning makes sense.
2. The method is clean and efficient for training.
3. The performance on different benchmarks compared to baseline methods is good.
4. It reduces the computational cost of generation compared to explicit CoT.

**Weaknesses:**

1. (minor) In Table 2, it would be beneficial to include the results for the thinking mode, which would make the table more comprehensive and the comparison more straightforward, allowing for a clearer view of the trade-off between computation and accuracy.

2. It would make the paper stronger if the authors could try larger models (such as 7B models) for LRT to show the method is scalable for larger base models. This should be doable (at least for SFT only) since the LRT method only needs to train the reasoning network, which can be smaller than the base model.

**Questions:**

What is $P\_{ref}$ in line 239 (since $P\_{\theta}$ is fixed during training) ?

---

> ### Author Response · Authors · 2025-11-21
> **Response to Reviewer dCpV**
>
> > 1. In Table 2, it would be beneficial to include the results for the thinking mode, which would make the table more comprehensive and the comparison more straightforward, allowing for a clearer view of the trade-off between computation and accuracy.
> > 2. It would make the paper stronger if the authors could try larger models (such as 7B models) for LRT to show the method is scalable for larger base models. This should be doable (at least for SFT only) since the LRT method only needs to train the reasoning network, which can be smaller than the base model.
>
> **Response:** We sincerely thank the reviewer for the positive assessment and the constructive suggestions regarding (i) including the results for the thinking mode and (ii) evaluating LRT on larger base models.
>
> Specifically, **we extended our LRT framework to the larger Qwen3-8B base model** and compared it against both the non-thinking and thinking modes. To ensure a comprehensive comparison of the computation-accuracy trade-off, we evaluated the thinking mode under token budgets of **1024, 2048, and 4096**.
>
> The results are summarized in the table below:
>
> | Method                  | Token Budget | AMC       | MATH-500  | GSM8K     | GPQA      | LSAT      | **Average** |
> | :---------------------- | :----------- | :-------- | :-------- | :-------- | :-------- | :-------- | :---------- |
> | Qwen3-8B (non-thinking) | 1024         | 46.25     | 77.05     | 73.88     | 46.21     | 38.80     | 56.44       |
> | Qwen3-8B (thinking)     | 1024         | 48.50     | 71.30     | 90.52     | 40.15     | 29.57     | 56.01       |
> | Qwen3-8B (thinking)     | 2048         | 51.25     | 83.50     | 90.54     | 48.23     | 39.67     | 62.64       |
> | Qwen3-8B (thinking)     | 4096         | 55.75     | 90.10     | 91.04     | 56.82     | 54.35     | 69.61       |
> | **Qwen3-8B (LRT)**      | 1024         | **50.25** | **78.20** | **91.02** | **44.43** | **34.26** | **59.63**   |
>
> **Analysis of the results:**
>
> 1.  **Scalability:** The results confirm that LRT scales effectively to larger LLMs. On the 8B backbone, LRT achieves an average score of **59.63**, clearly outperforming the non-thinking baseline (**56.44**). This demonstrates that our method is not limited to 1.7B/4B models and can effectively enhance the reasoning capabilities of larger, 8B-scale base models.
> 2.  **Computation-Accuracy Trade-off:** Including the thinking mode makes the trade-off much more transparent:
>     *   **LRT vs. Short Thinking (1024 tokens):** Under a restricted budget of 1024 tokens, LRT achieves a significantly higher average score (**59.63**) compared to the thinking mode (**56.01**). This highlights the efficiency of LRT in compressing reasoning information into latent representations.
>     *   **LRT vs. Long Thinking (2048+ tokens):** While the thinking mode with larger budgets (2048+ tokens) can further improve accuracy, it does so at the cost of significantly increased inference time and compute. Although explicit long-chain reasoning is beneficial for extremely hard questions, LRT offers a superior "sweet spot" for scenarios where low latency and efficiency are prioritized.
>
> We will include these Qwen3-8B experiments and the extended table including the thinking mode into the revised version, which we believe substantially strengthens the paper’s claims on both scalability and the efficiency–accuracy trade-off.

---

> ### Author Response · Authors · 2025-11-21
> **Response to Reviewer dCpV**
>
> **Question**: What is $P_{ref}$ in line 239.
>
> **Response:**$P_{\text{ref}}$ denotes the reference model used in RL algorithms (such as PPO or GRPO) to calculate the KL divergence penalty. It serves to regularize the policy model by preventing it from deviating too far from the initial distribution, thereby mitigating reward hacking. In our LRT framework, $P_{\text{ref}}$ specifically refers to the reasoning network initialized from the SFT stage, whose parameters remain frozen (fixed) during the RL training process.

---

> ### Author Response · Authors · 2025-11-24
> **Inquiry Regarding Rebuttal Feedback**
>
> Dear Reviewer  dCpV,
>
> We sincerely appreciate the time and effort you have dedicated to reviewing our manuscript. We provided detailed responses to your comments a few days ago and wanted to check if you had any further questions. We remain fully available to address any additional concerns you may have.

---

### Official Review · Reviewer_nNzk · 2025-11-01

**Soundness:** 2
**Presentation:** 2
**Contribution:** 2
**Rating:** 4
**Confidence:** 3

**Summary:**

This paper introduces the Latent Reasoning Tuning framework, a novel approach to enhance the reasoning capabilities of Large Language Models (LLMs) without modifying their parameters. The method involves training a lightweight, external reasoning network (specifically, Qwen3-Embedding-0.6B) to generate a set of latent vectors (256 learned embeddings) that condition the frozen base LLM's output. The framework is trained in two stages: Supervised Fine-Tuning (SFT) on the OpenR1-Math-220k dataset, followed by Reinforcement Learning (RL) fine-tuning on the DeepScaleR-Preview-Dataset.
The authors evaluate their method on several math and general reasoning benchmarks, including MATH and GPQA. The results demonstrate that this approach consistently outperforms the base models (tested on 1.7B and 4B parameter models) as well as other efficient reasoning baselines.

**Strengths:**

1. The paper addresses the critical and practical problem of improving the inference efficiency of LLM reasoning.
2. A key strength is the framework's efficiency. By only training a small reasoning network and keeping the base LLM's weights frozen, the method is computationally lightweight. This modular design also suggests high flexibility, as the reasoning network could potentially be paired with various pre-trained base models.
3. The authors provide ablation studies to justify key design choices, such as the necessity of the two-stage training pipeline (SFT followed by RL).
4. The method achieves consistent improvements over the selected baselines across multiple benchmarks, validating the effectiveness of the proposed latent reasoning approach.

**Weaknesses:**

1. One of the major drawback of the proposed method is the loss of interpretability of the reasoning traces. It would be great to provide some analysis on the learned reasoning vectors.
2. Although results show consistent improvements on 1.7B and 4B models, it remains unclear how the reasoning network scale with base model sizes. e.g. would a 0.6B reasoning backbone still sufficient for much bigger LLMs? Table 3 shows that the performance degrade when the number of reasoning token increases from 256 to 512, which seems to indicate that there's a sweet spot for the number of tokens depending on the model capacities, but this needs further experiments with various model sizes to verify.
3. The paper lacks an appendix and fails to provide crucial details about the hyperparameters used for the experiments. This omission significantly hinders the reproducibility of the work.
4. Although the method avoids the cost of fine-tuning the base LLM, it introduces the training and inference overhead of an additional reasoning model. The paper would benefit from a more direct comparison of the trade-offs (e.g., total training flops, inference latency) against other latent-variable baselines that do involve fine-tuning the base LLM, which would provide a more complete picture of the method's efficiency.

**Questions:**

1. How interpretable are the reasoning vectors?
2. Could you provide some insights on the scalability of LRT?

---

> ### Author Response · Authors · 2025-11-21
> **Response to Reviewer nNzk (1/4)**
>
> > 1 One of the major drawback of the proposed method is the loss of interpretability of the reasoning traces. It would be great to provide some analysis on the learned reasoning vectors.
>
> **Response:**
> We agree that replacing explicit, human-readable reasoning traces with latent vectors introduces an interpretability trade-off, and we appreciate the suggestion to analyze what the learned reasoning vectors represent. Below, we clarify their functional interpretation and provide an empirical analysis of their structure.
>
> **1. Theoretical Interpretation: Latent Vectors as Semantic Instructions**
> While the reasoning vectors are not linguistically interpretable like Chain-of-Thought, they are **functionally interpretable** based on our training objective.
> As formulated in Eq. (3) and Eq. (4) of our Method section, the reasoning network $G_\phi$ maps the input $X$ to latent vectors $z$, which are optimized to maximize the likelihood of the correct answer $Y$ from the frozen base LLM: $-\log P_{\theta}(Y \mid [X, z])$.
> Since these vectors are trained to substitute the explicit reasoning trajectory $R$ (where typically $P(Y|[X, R])$ yields the answer), the latent vectors $z$ can be interpreted as **compressed semantic instructions**. They encode the necessary intermediate "steering" information—such as task decomposition or key step identification—required by the base LLM to deduce the correct answer, stripped of the linguistic redundancy found in natural language.
>
> **2. Empirical Analysis: Visualization of Reasoning Patterns**
> To examine what these vectors capture in practice, we analyze the geometry of the latent space across different benchmarks. For each benchmark dataset (AMC, Math500, GSM8K, GPQA, LSAT), we first average its latent tokens over the sequence dimension to obtain a question-level latent vector. We then center these vectors by subtracting the global mean (over all questions from all datasets) and compute the average pairwise cosine similarity between questions from any two benchmarks.
>
> The results are presented in the table below:
>
> |         | **AMC** | **Math500** | **GSM8K** | **GPQA** | **LSAT** |
> | :------ | :-----: | :---------: | :-------: | :------: | :------: |
> | AMC     |  0.438  |    0.565    |  -0.173   |  0.104   |  -0.276  |
> | Math500 |  0.565  |    0.730    |  -0.223   |  0.141   |  -0.347  |
> | GSM8K   | -0.173  |   -0.223    |   0.076   |  -0.051  |  0.070   |
> | GPQA    |  0.104  |    0.141    |  -0.051   |  0.149   |  -0.032  |
> | LSAT    | -0.276  |   -0.347    |   0.070   |  -0.032  |  0.441   |
>
> **Observations:**
>
> 1.  **Domain Clustering:** Competition-style math benchmarks (AMC and Math500) exhibit the highest cross-dataset similarity ($\approx 0.565$) and strong self-similarity (e.g., Math500 $\approx 0.730$), indicating that the latent codes for these distinct but related math domains are closely aligned.
> 2.  **Semantic Separation:** In contrast, LSAT (logic/reading) shows positive within-domain similarity ($\approx 0.441$) but significant **negative similarity** to AMC and Math500 ($-0.276$ and $-0.347$, respectively). This suggests that the reasoning latents for logic-style problems occupy a distinctly different (and opposing) region of the latent space compared to olympiad-style math reasoning.
> 3.  **Complexity Stratification:** GSM8K and GPQA lie in between: they show moderate within-dataset similarity and relatively small cosine similarity with both competition math and LSAT, reflecting their mixed nature.
>
> These patterns indicate that the learned latent vectors are organized by **problem domain and difficulty**, effectively acting as compressed, task-specific instructions that guide the base LLM.

---

> ### Author Response · Authors · 2025-11-21
> **Response to Reviewer nNzk (2/4)**
>
> > 2 Although results show consistent improvements on 1.7B and 4B models, it remains unclear how the reasoning network scale with base model sizes. e.g. would a 0.6B reasoning backbone still sufficient for much bigger LLMs? Table 3 shows that the performance degrade when the number of reasoning token increases from 256 to 512, which seems to indicate that there's a sweet spot for the number of tokens depending on the model capacities, but this needs further experiments with various model sizes to verify.
>
> **Response:**
> We appreciate your insightful question regarding the scalability of LRT. In short, our additional experiments show that (i) a fixed 0.6B reasoning backbone scales well with larger base models (up to at least 8B), and (ii) there is indeed a capacity-dependent “sweet spot” in latent length: smaller bases saturate at shorter latent trajectories, whereas larger bases can still benefit from longer ones and are expected to peak at a larger length.
>
> **1. Scalability with base model size.**
> To examine scalability, we augment the original 1.7B and 4B results with an additional experiment on Qwen3-8B. As shown in the table below, performance continues to improve as we increase the base model size, while keeping the reasoning network fixed at 0.6B. This indicates that **the lightweight 0.6B reasoning network remains sufficient to guide larger models (at least up to 8B) without becoming a bottleneck.**
>
> | Base Model   |    AMC    | MATH-500  |   GSM8K   |   GPQA    |   LSAT    | **Average** |
> | :----------- | :-------: | :-------: | :-------: | :-------: | :-------: | :---------: |
> | Qwen3-1.7B   |   44.50   |   60.90   |   77.01   |   32.07   |   27.61   |    48.42    |
> | Qwen3-4B     |   46.25   |   72.60   |   88.51   |   39.27   |   28.59   |    55.04    |
> | **Qwen3-8B** | **50.25** | **78.20** | **91.02** | **44.43** | **34.26** |  **59.63**  |
>
> **2. Scalability with latent token length.**
>
> Regarding the “sweet spot” in the number of latent tokens, the reviewer’s intuition is correct and can be understood as a capacity–length trade-off. For the 1.7B base model (Table 3), increasing the latent length from 256 to 512 leads to a slight performance degradation, suggesting that 256 tokens already saturate the model’s effective capacity.
>
> For the 8B base model, we compare 256 vs. 512 latent tokens:
>
> | Latent Tokens |    AMC    | MATH-500  |   GSM8K   |   GPQA    |   LSAT    | **Average** |
> | :-----------: | :-------: | :-------: | :-------: | :-------: | :-------: | :---------: |
> |      256      |   50.25   |   78.20   |   91.02   |   44.43   | **34.26** |    59.63    |
> |      512      | **50.75** | **78.50** | **92.49** | **44.95** |   33.15   |  **59.97**  |
>
> As shown in the table above, **the 8B model still benefits from longer latent trajectories** in this range: going from 256 to 512 improves four out of five benchmarks and slightly increases the overall average.
>
> These results support the view that **the optimal number of latent tokens depends on the capacity of the base model**:
>
> - For smaller models (e.g., 1.7B), 256 tokens already saturate the model’s ability to effectively utilize additional latent information, so further increasing the length introduces redundancy and a small drop in performance.
> - For larger models (e.g., 8B), the base model has enough capacity to exploit richer latent information, allowing the same 0.6B reasoner to take advantage of longer latent sequences (512 tokens) and yield better performance; the performance–length curve is therefore expected to peak at a larger latent length.
>
> We will incorporate these new results and clarify this capacity–length trade-off in the revised version.

---

> ### Author Response · Authors · 2025-11-21
> **Response to Reviewer nNzk (3/4)**
>
> > 3 The paper lacks an appendix and fails to provide crucial details about the hyperparameters used for the experiments. This omission significantly hinders the reproducibility of the work.
>
> **Response:**
> We thank the reviewer for pointing this out and sincerely apologize for omitting detailed training configurations in the initial submission. We fully agree that these details are essential for reproducibility, and we will add a dedicated appendix with complete hyperparameter tables and training configurations in the revised version.
>
> Our framework is implemented using the open-source TRL[1] library. In the **SFT stage**, we train the reasoning network for 3 epochs with a batch size of 64 and a learning rate of $1 \times 10^{-3}$. In the subsequent **RL stage**, we use a batch size of 1024, generating 8 rollouts per question with a maximum rollout length of 2048 tokens. RL training is run for 100 steps with a learning rate of $1 \times 10^{-5}$ and a KL loss coefficient of $2 \times 10^{-3}$.
>
> [1] TRL: Transformer Reinforcement Learning. https://github.com/huggingface/trl

---

> ### Author Response · Authors · 2025-11-21
> **Response to Reviewer nNzk (4/4)**
>
> > 4 Although the method avoids the cost of fine-tuning the base LLM, it introduces the training and inference overhead of an additional reasoning model. The paper would benefit from a more direct comparison of the trade-offs (e.g., total training flops, inference latency) against other latent-variable baselines that do involve fine-tuning the base LLM, which would provide a more complete picture of the method's efficiency.
>
> **Response:**
> We appreciate the reviewer’s focus on efficiency trade-offs. The concern about the overhead of the additional reasoning module is valid. Most latent-variable baselines (e.g., recurrent hidden-state models) are still embedded in the standard autoregressive loop and thus have similar per-token speed as conventional generation. In contrast, LRT replaces the long explicit reasoning chain with a single forward pass through the 0.6B reasoner and a short answer generation. Therefore, we compare LRT against the two most relevant modes of the same base model: Qwen3 thinking and Qwen3 non-thinking.
>
> We measure average inference latency and throughput on 64 random MATH-500 problems using the Qwen3-1.7B base model:
>
> | Method                    | Latency (sec/question) | Throughput (tokens/sec) |
> | :------------------------ | :--------------------: | :---------------------: |
> | Qwen3-1.7B (thinking)     |         71.09          |          40.53          |
> | Qwen3-1.7B (non-thinking) |         14.62          |          48.93          |
> | **Qwen3-1.7B (LRT)**      |       **11.79**        |   **51.31 (73.02†)**    |
>
> * As shown above, **LRT achieves the lowest latency (11.79s)**, significantly faster than the Thinking mode (71.09s) and even outperforming the Non-thinking mode (14.62s). Although LRT introduces a small overhead from the 0.6B reasoning network, it guides the model to generate **more concise and direct answers**, reducing the total number of generated tokens compared to the non-thinking mode.
> * **LRT also exhibits the highest effective throughput**. The value 51.31 is the standard token-level throughput for the model’s output tokens; the value **73.02†** additionally counts the learned latent vectors as “reasoning steps”. This illustrates that LRT achieves a much higher information density per unit time than standard text-only generation.
>
> We will include these latency and throughput results in the revised version to provide a complete picture of the trade-offs.

---

> ### Author Response · Authors · 2025-11-24
> **Inquiry Regarding Rebuttal Feedback**
>
> Dear Reviewer nNzk,
>
> We sincerely appreciate the time and effort you have dedicated to reviewing our manuscript. We provided detailed responses to your comments a few days ago and wanted to check if you had any further questions. We remain fully available to address any additional concerns you may have.

---

> ### Author Response · Authors · 2025-11-26
>
> Dear Reviewer nNzk,
>
> We would like to once again express our sincere thanks for the time and effort you have devoted to reviewing our manuscript and for your thoughtful comments. We understand this is a particularly busy period with your own submissions and other responsibilities, and we truly appreciate the attention you have already given to our work.
>
> We are very excited about this paper and its findings, and we greatly value your perspective. If you have any further feedback or questions, or if anything in our responses could benefit from additional clarification, we would be very grateful to hear your thoughts during the discussion period and would be more than happy to elaborate.

---

> > ### Comment · Reviewer_nNzk · 2025-11-28
> >
> > Thank you for the thorough responses. My main concerns on the scalability and reproducibility have been addressed. I will raise my score accordingly.

---

> ### Author Response · Authors · 2025-11-28
>
> We sincerely thank the reviewer for the thoughtful follow-up and for taking the time to re-evaluate our submission. We are very glad to hear that the additional experiments and clarifications addressed your concerns regarding scalability and reproducibility. We will also release our source code to provide further implementation details and facilitate reproducibility.

---

> > ### Author Response · Authors · 2025-11-29
> >
> > To further support reproducibility, we have released an anonymized repository containing our source code, training and evaluation scripts, configuration files, and detailed instructions for reproducing our main results:
> > https://anonymous.4open.science/r/LatentReasoningTuning/

---

### Author Response · Authors · 2025-12-03
**Summary of Our Rebuttal**

Dear Area Chair,

Thank you for taking over during this unusual period. You are currently viewing our paper with the pre-rebuttal scores. We would like to briefly clarify and highlight what has changed. **After we submitted all rebuttals—but before the recent incident—Reviewers RA4D and gaFX had already raised their scores. Subsequently, Reviewer nNzk also confirmed a positive assessment and increased the final score.** All the changes can be referred in the discussion and timeline.

Below, we summarize our revisions and rebuttals, and explain how these updates address the primary concerns.

### Summary of Our Strengths

All four reviewers considered the reasoning-efficiency problem important, and several highlighted the novelty and practical value of our latent reasoning approach. Specifically:

**1. Novelty & Effectiveness.**
Reviewers described LRT as **novel, clean, and efficient** (nNzk, dCpV) and noted **consistent improvements across benchmarks** (nNzk, dCpV, RA4D).

**2. Modularity & Efficiency.**
They further emphasized LRT’s **modular design**—freezing the base LLM and training only a lightweight reasoning module—as a key source of **flexibility and computational efficiency** (nNzk, RA4D).

**3. Robustness of Motivation.**
Reviewer RA4D valued the trajectory-fragmentation analysis, which shows that models are **robust to token omission, thereby supporting the motivation for latent reasoning**.

### How Reviewer Concerns Are Addressed

**1. Scalability to Larger Base Models** (nNzk, dCpV, gaFX)

- We added extensive scaling experiments using Qwen3-8B (revised in Appendix D.1). The results confirm that **LRT scales effectively to larger base models**. Further discussion can be found in our responses to Reviewer nNzk (2/4), Reviewer dCpV, and Reviewer gaFX (3/3).

- We further conducted a controlled study on how the latent-token count interacts with base-model capacity (revised in Appendix D.2). Detailed discussion is provided in our response to Reviewer nNzk (2/4).

- **Both reviewers nNzk and gaFX confirmed that their scalability-related concerns regarding larger base models were addressed in our rebuttal.**

**2. Efficiency Analysis: Latency, Throughput, and Memory Usage** (nNzk, RA4D)

- We provide a complete efficiency analysis (revised in Appendix D.3), including inference latency, throughput, and peak memory usage. The results show that **LRT achieves the lowest latency, the highest effective throughput, and lower peak memory compared with the Qwen3 thinking mode**. Further discussion can be found in our responses to Reviewer nNzk (4/4) and Reviewer RA4D (2/4).

- **Reviewers nNzk and RA4D have stated that these efficiency-related issues were fully addressed in our rebuttal.**

**3. Interpretability of Latent Vectors and Relation to Prior Latent Reasoning Work** (nNzk, RA4D)

- We provide an empirical analysis of the latent representations (revised in Appendix D.4). **The results show that their geometric structure exhibits distinct clusters associated with different benchmarks**. Further details are provided in our response to Reviewer nNzk (1/4).

- Additionally, Appendix E presents an expanded discussion comparing LRT to recent latent reasoning approaches. Further details are given in our response to Reviewer RA4D (4/4).
- **Reviewer RA4D expressed that our rebuttal satisfactorily addressed their remaining concerns on these points.**

**4. More Implementation & Experimental Details.** (nNzk, RA4D, gaFX)

- Appendix B documents the full experimental configurations and hyperparameters. Further details are provided in our responses to Reviewer nNzk (3/4) and Reviewer gaFX (1/3).
- Appendix C provides a detailed description of the reasoning network architecture. Further details are provided in our response to Reviewer RA4D (1/4).
- Appendix D.5 presents our statistical significance analysis. Further details are provided in our response to Reviewer RA4D (3/4).
- Additional experiments with extended token budgets confirm that LRT maintains strong performance under longer reasoning horizons (see Table 1 in the revised manuscript). Further details are provided in our response to Reviewer gaFX (2/3).
- We have also released an anonymized code repository: https://anonymous.4open.science/r/LatentReasoningTuning/ (see Appendix B).
- **Reviewers nNzk, RA4D, and gaFX have confirmed that our rebuttal adequately addressed their concerns regarding experimental settings and implementation details.**

We sincerely thank all reviewers (nNzk, dCpV, RA4D, gaFX) for their thoughtful evaluations and for recognizing the novelty, motivation, and empirical contributions of our Latent Reasoning Tuning (LRT) framework. We have revised the manuscript accordingly and conducted sufficient experiments to resolve these issues thoroughly. We are also grateful to the Area Chair for the careful reading and the extra work required under these unusual circumstances.

Best regards,

The Authors

---

### Meta-Review · Area_Chair_PqqB · 2026-01-06

**Summary:**

The paper introduces Latent Reasoning Tuning, a novel framework that replaces explicit, token-by-token reasoning trajectories with compact latent representations. By utilizing a lightweight auxiliary reasoning network while keeping the base LLM frozen, the method significantly reduces inference latency compared to traditional Chain-of-Thought thinking modes. The reviewers reached a consensus that the approach is novel, efficient, and addresses a critical practical bottleneck in LLM reasoning.

**Reviewer Concerns:**

Addressed:
- The authors provided an exceptionally thorough rebuttal that successfully resolved all major technical concerns:
- New experiments on Qwen3-8B confirmed that the 0.6B reasoner scales effectively to larger base models.
- Geometric analysis of the latent space demonstrated that vectors cluster by domain and complexity, providing functional semantic instruction insights.
- Concrete metrics for latency, throughput, and memory usage were provided, proving the superior trade-off of LRT.
- The addition of a detailed appendix and an anonymized code repository satisfied concerns regarding implementation details.

Outstanding:
- LRT is optimized for high-throughput Fast Thinking. While explicit CoT remains stronger for extremely complex tasks, this is considered a deliberate trade-off. Since the base LLM is frozen, it retains a fallback for "Slow Thinking" when needed.

**Reviewer Scores:**

Three reviewers (nNzk, RA4D, gaFX) raised their scores after the rebuttal addressed scalability and technical gaps, joining a consistently positive dCpV.

---

### Decision · Program_Chairs · 2026-01-26

Accept (Poster)